# Identification and Characterization of the *Entamoeba Histolytica* Rab8a Binding Protein: A Cdc50 Homolog

**DOI:** 10.3390/ijms19123831

**Published:** 2018-11-30

**Authors:** Yuki Hanadate, Yumiko Saito-Nakano, Kumiko Nakada-Tsukui, Tomoyoshi Nozaki

**Affiliations:** 1Department of Parasitology, National Institute of Infectious Diseases, 1-23-1 Toyama, Shinjuku-ku, Tokyo 162-8640, Japan; yuki@nih.go.jp (Y.H.); yumiko@nih.go.jp (Y.S.-N.); kumiko@nih.go.jp (K.N.-T.); 2Graduate School of Life and Environmental Sciences, University of Tsukuba, 1-1-1 Tennodai, Tsukuba, Ibaraki 305-8572, Japan; 3Graduate School of Medicine, The University of Tokyo, 7-3-1 Hongo, Bunkyo, Tokyo 113-0033, Japan

**Keywords:** Rab8, Cdc50, endoplasmic reticulum, *Entamoeba*, protozoa, miltefosine

## Abstract

Membrane traffic plays a pivotal role in virulence in the enteric protozoan parasite *Entamoeba histolytica*. EhRab8A small GTPase is a key regulator of membrane traffic at the endoplasmic reticulum (ER) of this protist and is involved in the transport of plasma membrane proteins. Here we identified the binding proteins of EhRab8A. The Cdc50 homolog, a non-catalytic subunit of lipid flippase, was identified as an EhRab8A binding protein candidate by affinity coimmunoprecipitation. Binding of EhRab8A to EhCdc50 was also confirmed by reciprocal immunoprecipitation and blue-native polyacrylamide gel electrophoresis, the latter of which revealed an 87 kDa complex. Indirect immunofluorescence imaging with and without Triton X100 showed that endogenous EhCdc50 localized on the surface in the absence of permeabilizing agent but was observed on the intracellular structures and overlapped with the ER marker Bip when Triton X100 was used. Overexpression of *N*-terminal HA-tagged EhCdc50 impaired its translocation to the plasma membrane and caused its accumulation in the ER. As reported previously in other organisms, overexpression and accumulation of Cdc50 in the ER likely inhibited surface transport and function of the plasma membrane lipid flippase P4-ATPase. Interestingly, HA-EhCdc50-expressing trophozoites gained resistance to miltefosine, which is consistent with the prediction that HA-EhCdc50 overexpression caused its accumulation in the ER and mislocalization of the unidentified lipid flippase. Similarly, *EhRab8A* gene silenced trophozoites showed increased resistance to miltefosine, supporting EhRab8A-dependent transport of EhCdc50. This study demonstrated for the first time that EhRab8A mediates the transport of EhCdc50 and lipid flippase P4-ATPase from the ER to the plasma membrane.

## 1. Introduction

Membrane traffic is involved in a variety of cellular processes including transport of newly synthesized proteins, membrane compartmentalization and communication between organelles [1]. Rab GTPases are important regulators of intracellular membrane traffic, which is widely conserved in eukaryotes [1,2]. Together with Rab specific binding proteins, Rab GTPases regulate various trafficking steps including vesicle budding and cargo sorting from the donor membrane [3,4], vesicle transport along the cytoskeleton [5,6] and vesicle tethering and fusion with the acceptor membrane [7]. In each step, organelle-specific effectors and Rab binding proteins interact with Rab in the GTP-bound active or GDP-bound inactive form and maintain organelle-specific membrane dynamics [2].

Identification of binding proteins of Rab GTPases often provides necessary clues to the mechanisms of trafficking pathways [8]. A plethora of binding proteins and effectors for most Rab GTPases have been studied to elucidate the molecular mechanism of the Rab-mediated trafficking in human and major model organisms [2]. For example, Rab11 and Rab8 coordinately control trafficking from the trans-Golgi to the plasma membrane in mammalian cells together with their effector proteins, Rab11-family interacting proteins (FIPs) and Rabin8 (Rab8-guanine nucleotide exchange factor). Neurite outgrowth requires a cascade of Rab11, Rabin8 and Rab8 activities in the recycling endosomes [9]. In ciliary membrane trafficking, Rab11 and Rab8 sort rhodopsin at the trans-Golgi membrane and transport rhodopsin to the periciliary membrane of photoreceptor cells followed by the Golgi recruitment of FIP3 and Rabin8 [10]. In the social amoeba *Dictyostelium discoideum*, intracellular osmoregulation is controlled by contractile vacuole fusion with the plasma membrane under hypotonic conditions; and this process is regulated by the serial recruitment of Rab11, Rab11 effector, Rab8 and Rab8-GTPase activating protein on the contractile vacuole membrane [11]. Thus, the Rab11 and Rab8 cascade, which is mediated by Rab8 effector proteins, is conserved in many tissues and organisms.

The pathogenesis of the enteric protozoan parasite *Entamoeba histolytica* is tightly linked to membrane trafficking in a variety of manners including the transport of membrane receptors necessary for the attachment to host cells, secretion of virulence factor cysteine proteases for tissue destruction and trogocytosis of live immune and non-immune host cells [12,13]. Expansion of Rab genes in the *E. histolytica* genome reflects the importance of membrane traffic in this parasitic protist [14,15]. Among the more than 100 Rab genes encoded by the *E. histolytica* genome, only 10 EhRab proteins (EhRab1B/EhRab35, EhRab5, EhRab7A, EhRab7B, EhRab8A, EhRab11A, EhRab11B, EhRab21, EhRabA and EhRabB) have been functionally characterized [14,16,17,18]. However, effectors and binding proteins have not been identified for EhRab proteins except for EhRab7A [19]. Binding of the retromer complex, more specifically, one of its component Vps26, to Rab7 was first reported in *E. histolytica* [19] and subsequently observed in mammalian cells [20], indicating that Rab7 is involved in cargo recognition and sorting in the endosome of a wide range of eukaryotes.

We have previously shown that EhRab8A, an amebic homolog of human Rab8, is localized to the ER [17]. This localization is unique and in good contrast to Rab8 in humans, which is localized to the trans-Golgi and regulates transport to the plasma membrane in cooperation with Rab8 effectors known as FIP [21]. We proposed that EhRab8A may be involved in the transport of plasma membrane proteins, as suggested by gene-silencing experiments in which EhRab8A repression caused defects in adhesion [17]. Here, we report the identification and characterization of a putative EhRab8A binding protein, Cdc50, which is involved in the transport of a putative lipid flippase from the ER to the cell surface in *E. histolytica*.

## 2. Results

### 2.1. Identification of EhCdc50 As an EhRab8A Interacting Protein in E. histolytica

We previously reported that EhRab8A is localized to the ER in the steady state, which differs from the Rab8 ortholog in other organisms [17,22]. In mammals, Rab8 is localized to the trans-Golgi or endocytic compartments and involved in transport to the plasma membrane [17,22]. However, the function of the Golgi apparatus and the trans-Golgi in *Entamoeba* is highly diverse in protein glycosylation and organelle function [23]. Curiously, EhRab8A gene silencing demonstrated that EhRab8A is involved in the trafficking of at least three surface proteins with molecular masses of 200, 60 and 30 kDa [17]. In general, the protein sorting of secretory proteins in the ER is known to be regulated by the Sar1 GTPase but not by Rab GTPases [24]. To clarify the mechanisms of the EhRab8A-dependent trafficking across the ER, we attempted to identify EhRab8A interacting proteins by coimmunoprecipitation of the epitope-tagged EhRab8A. 

To see whether EhRab8A forms a stable complex with other proteins and, if so, to estimate its apparent molecular mass, we conducted BN-PAGE following immunoprecipitation (Figure 1a). Immunoblotting with the anti-Myc antibody revealed an 87-kDa band in the immunoprecipitated sample from lysates of *N*-terminal Myc-tagged EhRab8A-expressing cells (Figure 1a, arrowhead), suggestive of possible interacting protein(s). The putative EhRab8A interacting protein(s) that were coimmunoprecipitated with Myc-EhRab8A using an anti-Myc antibody were detected as two specific bands of approximately 35 and 40 kDa (Figure 1b, arrowheads). 

The two bands were excised from the gel and analyzed by liquid chromatography and time-of-flight tandem mass spectrometry (LC-ToF MS/MS) (Table A1). From the 35 kDa band, two candidate proteins, EHI_142740 and EHI_118780, were detected. EHI_142740 was detected exclusively from Myc-EhRab8A expressing cells but not from mock control cells (Table 1). This protein showed homology with human CDC50 (33% identity, e-value 1.3 × 10^−33^) and *Leishmania* Cdc50 homologue, Los3 (25% identity, e-value 3.0 × 10^−25^), which are known as a non-catalytic subunit of lipid flippase P4-ATPase. Thus, we designated hereinafter EHI_142740 as EhCdc50 (GenBank Accession number, LC389589). The other candidate detected in the 35 kDa band, EHI_118780, showed homology to a nuclear pore protein. 

From the 40 kDa band, six candidate proteins involved in lipid metabolism were identified: sphingomyelin phosphodiesterase (EHI_100080), glycerophosphodiester phosphodiesterase (EHI_068320), cytosolic tldc domain-containing protein (EHI_134660), C2 domain containing protein (EHI_069950), lysosomal vacuolar ATP synthase subunit (EHI_106350) and mitosome luminal protein sulfate adenylyltransferase (EHI_197160) (Table 2). Among these candidates, only glycerophosphodiester phosphodiesterase (EHI_068320) is known to be localized to the ER membrane in other organisms and thus considered to be a candidate of EhRab8A binding protein. However, the localization and function of the 40 kDa band as a potential EhRab8A interacting protein and the binding of these proteins to EhRab8A remain elusive, because we failed to establish amoeba transformants expressing EHI_068320 despite our repeated attempts.

### 2.2. Confirmation of Binding of EhCdc50 and EhRab8A

The interaction of EhCdc50 and EhRab8A was confirmed by reciprocal coimmunoprecipitation of EhRab8A from *N*-terminal HA-tagged-EhCdc50-expressing cells (Figure 1c). The immunoprecipitated HA-EhCdc50 was recognized as a 42.5 kDa band using an anti-HA antibody; the size was slightly larger than the calculated molecular mass of HA-EhCdc50 of 38.9 kDa. This is consistent with the prediction that Cdc50 is a glycosylated transmembrane protein (see below). EhRab8A was detected as a 23 kDa band with anti-EhRab8A antiserum in the coimmunoprecipitated sample, while the negative control marker for endosomal protein Vps26 [19], which was previously shown to be differentially localized from EhRab8A [17], was not detected in the sample. These data clearly validated the physical interaction between EhRab8A and EhCdc50. 

### 2.3. EhCdc50 is N-Glycosylated Protein

The Cdc50 proteins are structurally well conserved at the primary sequence levels throughout eukaryotes and contain two transmembrane domains separated by a large exoplasmic loop (Figure 3a, see below), which contains two to four *N*-glycosylation sites [Asn-Xaa-(Ser/Thr)] in *Arabidopsis thaliana*, *Saccharomyces cerevisiae* and human [25,26,27]. In human Cdc50A and *S. cerevisiae* Cdc50p, it has been demonstrated that post-translational modifications at the four *N*-glycosylation sites are necessary for the stable expression of lipid flippase P4-ATPase, complex formation and trafficking from the ER [26,27]. In *E. histolytica*, sequence-based prediction suggested that three of four *N*-glycosylation sites are present in the ectodomain (Asn141, Asn162 and Asn244) (Figure A1). Inhibition of *N*-glycosylation by tunicamycin treatment decreased the apparent molecular weight of HA-EhCdc50 from 42.5 kDa to 37.8 kDa, as shown by immunoblot analysis using an anti-HA antibody (Figure 2), indicating that EhCdc50 is modified with *N*-linked oligosaccharides. 

### 2.4. Localization of Endogenous EhCdc50 and Exogenous Epitope-Tagged EhCdc50 and Inhibition of Its Transport to the Plasma Membrane by Overexpression

Cdc50 is known as a non-catalytic subunit of lipid flippase P4-ATPase and the complex localizes to the endosomes and plasma membrane in the steady state [28,29]. Gene silencing of EhCdc50 was unsuccessful despite our repeated transfections, indicating EhCdc50 is essential for the growth. To investigate the localization of endogenous and exogenously overexpressed EhCdc50, we established a native antibody against recombinant EhCdc50_108-271_ (Figure 3a). EhCdc50 was detected on the cell surface in the mock transformant with this anti-EhCdc50 antibody (similar results were observed in wild type cells; data not shown) when trophozoites were not treated with Triton X-100, suggesting that endogenous EhCdc50 was uniformly distributed on the entire plasma membrane (Figure 3b). The surface labeling by anti-EhCdc50 antibody was significantly reduced when recombinant EhCdc50_108-271_ protein was used for competition; the relative intensity of the peripheral staining by anti-EhCdc50 antibody was reduced to 19 ± 12% of the level in the absence of recombinant EhCdc50_108-271_ protein by the preincubation of the anti-EhCdc50 antibody with recombinant EhCdc50_108-271_ protein (Figure 3b,c), validating specificity of this antibody against EhCdc50. We next examined the localization of EhCdc50 in HA-EhCdc50-expressing cells using an anti-EhCdc50 antibody. In HA-EhCdc50-expressing cells, EhCdc50 was not detected on the cell surface of the nonpermeabilized trophozoites with ani-EhCdc50 antibody (Figure 3b, right panel), suggesting that transport of both HA-EhCdc50 and native EhCdc50 to the plasma membrane was prevented. 

When HA-EhCdc50 cells were perforated with Triton X-100, HA-EhCdc50 appeared to be associated with the ER-like network structures (Figure 4a, upper panel), which was confirmed by co-staining with an anti-EhBip (ER luminal chaperone) antibody (Figure 4a). HA-EhCdc50 showed strong colocalization with EhBip (Pearson’s correlation coefficient: *R* = 0.65). HA-EhCdc50 also showed mild colocalization with EhRab8A, as visualized using anti-HA and anti-EhRab8A antibodies (*R* = 0.47) (Figure 4b). Colocalization of EhRab8A and EhBip was previously demonstrated with a Pearson’s correlation coefficient of 0.7 [17]. These results indicate that overexpressed HA-EhCdc50 is mainly localized to the ER and partially colocalized with EhRab8A. Similar observations suggesting that overexpressed Cdc50 was accumulated in the ER were reported in other organisms including in human and yeast [30,31,32].

### 2.5. Overexpression of EhCdc50 or Gene Silencing of EhRab8A Showed Decreased Miltefosine Susceptibility

Many of phospholipid flippase P4-ATPases require interactions with the Cdc50 family for their exit from the ER [30,32]. In *Arabidopsis*, *S. cerevisiae* and mammalian cells, either overexpression or knockout of Cdc50 caused accumulation of Cdc50 together with P4-ATPase to the ER, which led to defects in the internalization of phospholipids [30,31,32]. We hypothesized that EhCdc50 and a catalytic subunit of phospholipid flippase P4-ATPase on the plasma membrane are involved in translocation of phospholipids from the exoplasmic to the cytoplasmic face in *E. histolytica*. The effect on phospholipid translocation in EhCdc50-overexpressing cells was evaluated using the phosphocholine analogue miltefosine, which is toxic and causes amebic cell growth inhibition [33]. Miltefosine is an alkylated phosphocholine, originally developed as an anticancer drug and is effective against six different *Leishmania* species, including *L. donovani*, *L. aethippoca*, *L. tropica*, *L. mexicana*, *L. panamensis* and *L. major* [34]. It has been shown in *Leishmania* that the deletion of the Cdc50 homologue, LdRos3, confers resistance to miltefosine [35]. We examined the susceptibility of an HA-EhCdc50-expressing strain against miltefosine. After 18-h treatment with miltefosine, the cell viability of the HA-EhCdc50 expressing strain was higher than that of the control mock transformant (The IC_50_ values of mock and HA-EhCdc50-expressing strains against miltefosine, 22.2 ± 3.2 and 33.4 ± 4.4 µM, respectively; *p* value <0.05) (Figure 5a). Similarly, the EhRab8A gene-silenced strain showed partial resistance to miltefosine (The IC_50_ values of G3 parental and EhRab8A gene silenced strains were 21.8 ± 3.6 and 29.2 ± 2.8 µM, respectively; *p* value <0.05) (Figure 5b). 

## 3. Discussion

### 3.1. Identification of EhCdc50 as EhRab8A Binding Protein

Cdc50 is an essential accessory protein for the exit of lipid flippase P4-ATPase from the ER in yeast, *Leishmania*, human and plant [28,30,36]. In this study, we identified the non-catalytic subunit, EhCdc50, of lipid flippase as a binding protein of EhRab8A small GTPase. The interaction between EhRab8A and EhCdc50 was confirmed by reciprocal immunoprecipitation (Figure 1) and immunofluorescence assay (Figure 4) in the present study and appears to be a common feature conserved among eukaryotes. Rab8 appears to be highly conserved in eukaryotes including Opistokonta (Metazoa and fungi), Viridiplantae (*A. thaliana*) and Amoebozoa (*E. histolytica* and *Dictyostelium discoideum*) [37,38] but not in Trypanosomatidae (*Leishmania* and *Trypanosoma*) and some protozoan parasites that belong to the Alveolata (*Plasmodium* and *Toxoplasma*) [39,40]. However, EhRab8A is unique in that it resides in the ER [17], while in human macrophages and *Dictyostelium*, Rab8 homologues localize to phagosomes or endocytic compartments [10,11]. In *E. histolytica*, EhRab8A is involved in the transport of surface proteins involved in the cellular attachment and phagocytosis [17]. Thus, identification of Cdc50 as a binding protein of Rab small GTPase is plausible, as the flippase and thus its non-catalytic subunit, must be specifically transported via an EhRab8A-dependent secretory pathway, although this has never been demonstrated in other organisms.

EhRab8A gene-silencing decreased amebic adhesion to host cells and reduced the presentation of major three surface proteins with apparent molecular weights of 200, 60 and 30 kDa [17]. As intrinsic EhCdc50 has a calculated molecular mass of 35 kDa, it does not seem to correspond to one of the three previously identified major surface proteins. This explains why the HA-EhCdc50 expressing strain, in which HA-EhCdc50 is accumulated in the ER (Figure 4), did not show a defect in phagocytosis of erythrocytes (Figure A2). Taken together, EhRab8A is likely involved in the transport of multiple proteins including EhCdc50 as well as three previously reported major surface proteins from the ER to the cell surface.

### 3.2. N-Linked Oligosaccharide Modification and Possible Role of EhCdc50 in the Exit From the ER

It has been shown that in mammalian and yeast cells Cdc50 has an apparent molecular mass of 50–60 kDa, is glycosylated in the Golgi and presented on the cell surface [28]. In *Entamoeba*, EhCdc50 was detected as a band with an approximate molecular weight of 35 kDa by SDS-PAGE (Figure 1b), which is consistent with the estimated molecular mass of the polypeptide (35.7 kDa), suggesting that EhCdc50 is not glycosylated [41]. However, as EhCdc50 possesses two hydrophobic transmembrane domains, which are likely responsible for the aberrant mobility on SDS-PAGE and offsetting possible glycosylations. Indeed, tunicamycin treatment reduced the molecular mass of HA-EhCdc50 by approximately 4.7 kDa (Figure 2), indicating that some of the three potential *N*-glycosylation sites in EhCdc50 (Figure A1) may be glycosylated. This predicted addition of a relatively short sugar chain is likely explained by the fact that *E. histolytica* forms an unusual simple 1.8 kDa *N*-glycan precursor of Man_5_GlcNAc_2_ in the ER rather than the common 2.5 kDa Glc_3_Man_9_GlcNAc_2_, which is present in most animals, plants and fungi [41]. 

*N*-glycosylation of Cdc50 is required for stability and activity of the Cdc50/ATPase complex in mammalian cells [27]. It is currently unclear whether *N*-glycosylation of EhCdc50 is necessary for the interaction with EhRab8A. The results of BN-PAGE and immunoblotting showed that EhRab8A formed an 87 kDa complex (Figure 1a), suggesting that the ~110 kDa amoebic EhP4-ATPase, which is present as a family of 11 proteins encoded in the genome (see below), is not part of the EhRab8A/EhCdc50 complex. After EhRab8A recognizes EhCdc50 and completes its sorting in the ER, EhRab8A is presumably replaced by EhP4-ATPase and then the *N*-glycosylated EhCdc50/EhP4-ATPase complex can be transported to the plasma membrane.

### 3.3. A Possible Novel Rab8A-Dependent and COPII-Independent Pathway for Cdc50 Traffic

One of the assumed roles of Cdc50 is trafficking of P4-ATPase from the ER to the cell surface. As suggested by a previous proteome analysis of the COPII coat component in yeast, P4-ATPase Nep1p physically interacts with the COPII coat components Sec13p, Sec23p and Sec24p [42]. It has been also shown that yeast P4-ATPase Drs2p interacts with Sec23p [42]. Analogously, EhCdc50 appears to be transported in a similar manner as Sar1p GTPase and COPII. However, EhRab8A does not colocalize with the COPII component on the ER [17], indicating the presence of an unconventional COPII-independent and EhRab8A-dependent ER exit pathway in *Entamoeba*. 

It has been reported in many organisms that both repression and overexpression of Cdc50 and/or P4-ATPase prevents correct targeting of the P4-ATPase/Cdc50 complex and results in accumulation of the complex in the ER, concomitant with the loss of the plasma membrane-associated lipid flippase activity [31,32,43]. With a few exceptions, the class 2 P4-ATPases, yeast P4-ATPase Neo1p and mammalian ATP9A and ATP9B, do not require the Cdc50 family. Yeast Neo1p forms a complex with the scaffolding protein Ysl2p [44] and mammalian ATP9A and ATP9B exit the ER to localize with the respective cell membranes in the absence of Cdc50 [45]. We observed the plasma membrane localization of EhCdc50 in the steady state (Figure 3). In contrast to the single gene copy of EhCdc50, we found 11 potential P4-ATPase genes in the *E. histolytica* genome: EHI_141350, EHI_096620, EHI_135220, EHI_188210, EHI_174280, EHI_049640, EHI_197300, EHI_168260, EHI_024120, EHI_140130 and EHI_009460. Three of them, EHI_049640, EHI_168260 and EHI_197300, were classified as class 2 P4-ATPases (our unpublished results), suggesting that the EhCdc50 and P4-ATPase in the amoebic genome and several P4-ATPases exit the ER via an EhCdc50-independent mechanism.

### 3.4. EhRab8A and EhCdc50 Are Involved in Miltefosine Sensitivity in Entamoeba

Miltefosine is known to inhibit choline-phosohate cytidylyltransferase involved in the phosphatidylcholine biosynthesis in mammalian cells, [46] and homologous proteins are present in *Leishmania* [47]. However, the leishmanicidal mode of action on miltefosine is not completely understood and most information regarding its properties was obtained from miltefosine-resistant parasites generated in vitro and clinical isolates. Miltefosine resistance in *Leishmania* occurs mainly through the reduction in drug incorporation associated with the introduction of mutations into the P4-ATPase miltefosine transporter, MT and Cdc50 homologue, LdLos3 [48]. The resistance mechanism is likely conserved among *Leishmania* species causing either cutaneous (*L. amazonensis* and *L. major*) or visceral (*L. donovani*) leishmaniasis [49,50]. Here we showed that overexpression of HA-EhCdc50 caused defects in EhCdc50/P4-ATPase trafficking and a concomitant increase in the tolerance to miltefosine (Figure 5a), suggesting that EhCdc50 is involved in the transport of phospholipids and miltefosine by P4-ATPase. Additionally, down-regulation of EhRab8A similarly caused tolerance to miltefosine (Figure 5b), supporting the prediction that EhRab8A is necessary for the transport of EhCdc50/P4-ATPase. Our result indicates that any mutations in EhRab8A that affect the affinity with EhCdc50 or an increase in the expression of EhCdc50 can confer resistance to miltefosine in *Entamoeba*. Thus, EhRab8A and EhCdc50 are of interest as potential mechanisms of drug resistance against miltefosine in case it is used in clinical cases in future. In the human fungal pathogen *Cryptococcus*, it is reported that Cdc50 is involved in its virulence as well as in drug resistance [51].

The mechanisms of EhCdc50/P4-ATPase in phospholipid transport and the EhRab8A-mediated regulation of EhCdc50/P4-ATPase trafficking remain unclear in *E. histolytica*. *E. histolytica* possesses full capacity of de novo biosynthesis of phospholipids including phosphatidylcholine and phosphatidylethanolamine [52] and the ability to scavenge components from the environment under in vitro cultivation conditions [53]. Furthermore, *Entamoeba* possesses a list of lipid binding proteins in their genome [54,55]; one of such lipid transfer proteins, EhPCTP-L, was shown to bind phosphatidylserine and phosphatidic acid and localizes to the cell surface [56]. The catalytic P4-ATPase subunit of the phospholipid flipase responsible for miltefosine transport should be identified to better understand the molecular mechanisms of phospholipid transport on the plasma membrane and its involvement in drug resistant and pathogenesis in *Entamoeba*.

## 4. Materials and Methods 

### 4.1. Culture

Trophozoites of *E. histolytica* HM-1: IMSS cl6 (HM-1) [57] and G3 [58] strains were cultured axenically at 35.5 °C in 13 × 100 mm screw-capped Pyrex glass tubes or 25 cm^2^ plastic culture flasks with BI-S-33 medium, as previously described [53,57]. 

### 4.2. Antibodies

The anti-EhBip [17], anti-EhRab8A [17], anti-EhVps26 [19] antibodies used were described previously. The anti-HA antibody clone 16B12 and anti-Myc clone 9E10 monoclonal antibodies were purchased from Santa Cruz Biotechnology (Dallas, TX, USA). Anti-EhCdc50 antibody was produced as described below.

### 4.3. Creation of E. histolytica Transformant Lines

A plasmid expressing *N*-terminal 3HA-tagged EhCdc50 (HA-EhCdc50) was constructed as follows. A 966-base pair DNA fragment containing the EhCDC50-coding sequence was amplified by PCR from *E. histolytica* genomic DNA using the following oligonucleotides: 5′-CCCGGGATGTCAGAGAAAGTAAAGGGACTTG-3′ and 5′-CTCGAGTTACCATCGAAGAAATCTC-3′ (the SmaI and XhoI restriction sites are underlined). The amplified fragment was cloned into pCR™-Blunt II-TOPO™ with a Zero Blunt™ TOPO™ PCR Cloning Kit (Invitrogen, Carlsbad, CA, USA). This plasmid was digested with SmaI and XhoI and the insert was cloned into SmaI- and XhoI-digested pKT-3HA. The plasmid was introduced into trophozoites as described previously [59]. Construction of the 3Myc-EhRab8A (Myc-EhRab8A)-expressing line from HM-1 and EhRab8A gene-silenced line from G3 strain were described previously [17].

### 4.4. Blue-Native Polyacrylamide gel Electrophoresis (BN-PAGE)

Approximately 1 × 10^6^ cells were harvested at the late logarithmic growth phase, washed with cold phosphate-buffered saline (PBS) and homogenized in 0.5 mL homogenization buffer (250 mM sucrose, 50 mM Tris-HCl, 50 mM NaCl, 5 mM MgCl_2_, 1 mM E-64, pH 7.5) with 30 strokes of a Dounce homogenizer. Samples were centrifuged at 13,000 *g* at 4 °C for 10 min to obtain the pellet fraction (p13). The p13 fraction was solubilized in homogenization buffer containing 2% digitonin at 4 °C for 30 min. Solubilized samples were collected by centrifugation (13,000 *g*, 4°C, 10 min) and subjected to BN-PAGE using the NativePAGE^TM^ NovexH Bis-Tris Gel System (Invitrogen) according to the manufacturer’s protocol. The resolved proteins were transferred to a polyvinylidene fluoride membrane and the Myc-EhRab8A protein complex was detected using anti-Myc antibody.

### 4.5. Coimmunoprecipitation

Approximately 4 × 10^6^ Myc-EhRab8A-expressing or mock control trophozoites were harvested cells in the logarithmic growth phase, washed with 2% glucose/PBS and resuspended in 1 mL of PBS. The cell suspension was incubated with 2 mM 3,3′-dithiodipropionic acid di(*N*-hydroxysuccinimide ester) (DSP) (Sigma-Aldrich, St. Louis, MO, USA) for cross-linking for 30 min at room temperature according to the manufacturer’s protocol. The samples were mixed with 1 mL homogenization buffer (250 mM sucrose, 50 mM Tris-HCl, 50 mM NaCl, 5 mM MgCl_2_, 1 mM E-64, pH 7.5) and homogenized on ice with 30 strokes of a Dounce homogenizer. After unbroken cells were removed by centrifugation at 400 g for 2 min, the supernatant was centrifuged to obtain the p13 fraction which was primarily enriched with EhRab8A [17]. The p13 fraction was resuspended in approximately 0.5 mL of homogenization buffer containing 2% digitonin at a 2 mg/mL protein concentration and then incubated with Protein G Sepharose (Sigma-Aldrich) at 4 °C for 60 min to reduce non-specific binding during coimmunoprecipitation. The Myc-EhRab8A protein complex was immunoprecipitated with anti-Myc-antibody-conjugated agarose (Sigma-Aldrich) at 4 °C for 3 h and then washed with homogenization buffer containing 0.5% digitonin. The Myc-EhRab8A protein complex was eluted with homogenization buffer containing 2% digitonin and 0.4 mM Myc peptide (Sigma-Aldrich). 

### 4.6. Liquid Chromatography-Tandem Mass Spectrometric Analysis (LC-MS/MS)

In-gel trypsin digestion of protein bands of interest and LC-MS/MS mass spectrometric analysis (Orbitrap, Thermo Fisher Scientific, Waltham, MA, USA) were performed as previously described [60,61].

### 4.7. Production of EhCdc50 Antibody

The exoplasmic region of EhCdc50, corresponding to amino acid residues 108-271, was inserted into the pCold-GST vector (Takara Bio, Shiga, Japan). EhCdc50_108-271_ recombinant protein was expressed in the *Escherichia coli* BL21 (DE3) strain (Thermo Fisher Scientific) and purified according to the manufacturer’s instructions. Anti- EhCdc50 antiserum was commercially raised against recombinant EhCdc50_108-271_ in rabbits (Eurofins Genomics, Val Fleuri, Luxembourg). 

### 4.8. Indirect Immunofluorescence Assay

An indirect immunofluorescence assay was conducted essentially as previously described [62]. Trophozoites were seeded into 8-mm round wells on a glass slide, fixed with 3.7% paraformaldehyde, permeabilized with 0.1% Triton X-100 and reacted with antibodies. Alexa 488- or Alexa 568-conjugated anti-mouse or rabbit IgG (Molecular Probes, Eugene, OR, USA) were used as secondary antibodies. Images were acquired using an LSM780 confocal laser-scanning microscope (Zeiss, Oberkochen, Germany). Images were analyzed using Zeiss ZEN software.

### 4.9. Miltefosine Sensitivity Assay

Trophozoites were seeded into 96-well plates at 0.5 × 10^4^ cells/well in 280 µL of BI-S-33 medium containing 5–80 µM Miltefosine. After 18 h of culture, the medium was removed and the viability of attached trophozoites was estimated by measuring of the absorbance at 450 nm using a DTX880 Multimode Detector (Beckman Coulter, Brea, CA, USA) after incubating of the trophozoites with 100 µL of Opti-MEM medium containing 10% WST-1 reagent (Roche, Basel, Switzerland) for 20 min at 37 °C [63]. Experiments were repeated three times with triplicate samples evaluated in each experiment.

## 5. Conclusions

We identified Cdc50 homologue as a binding protein of EhRab8A in *E. histolytica*. Our data suggested that EhRab8A mediated the transport of EhCdc50 and lipid flippase P4-ATPase from the ER to the plasma membrane.

## Figures and Tables

**Figure 1 ijms-19-03831-f001:**
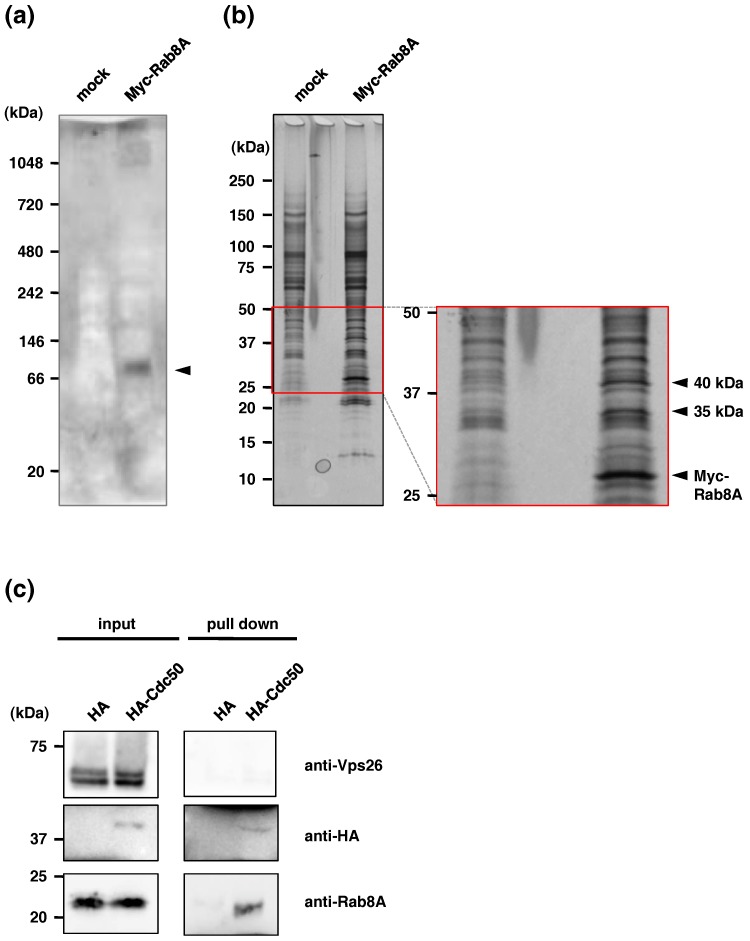
Identification and confirmation of EhCdc50 as an EhRab8A binding protein. (**a**) Immunoblot analysis of Myc-EhRab8A with anti-Myc antibody following BN-PAGE from representative image of three independent experiments. (**b**) SDS-PAGE analysis of Myc-EhRab8A-binding proteins. Myc-EhRab8A-binding proteins coimmunoprecipitated with anti-Myc antibody were separated on SDS-PAGE and detected with silver staining. Representative data of three independent experiments was shown. (**c**) Reciprocal coimmunoprecipitation of EhRab8A via interaction with EhCdc50. EhCdc50 binding protein was immunoprecipitated from lysates of HA-EhCdc50-expressing cells with anti-HA antibody, followed by immunoblotting with anti-HA, anti-EhRab8A and anti-EhVps26 antibodies. Anti-EhVps26 antibody was used as a negative control.

**Figure 2 ijms-19-03831-f002:**
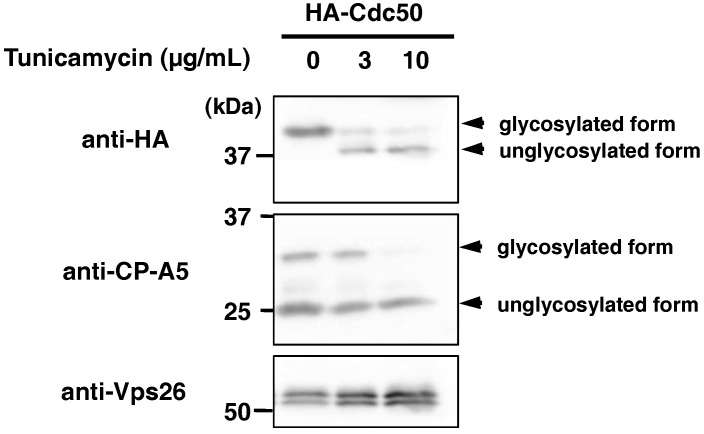
Demonstration of *N*-linked glycosylation on EhCdc50. HA-EhCdc50 expressing cells were treated with tunicamycin at 1, 3, 10 µg/mL for 24 h. Cells were lysed and analyzed by immunoblotting with anti-HA, anti-EhCP-A5 and anti-EhVps26 antibodies. The apparent molecular weight of HA-EhCdc50 was decreased by the tunicamycin treatment. EhCP-A5 and EhVps26 are the glycosylated and non-glycosylated proteins, respectively.

**Figure 3 ijms-19-03831-f003:**
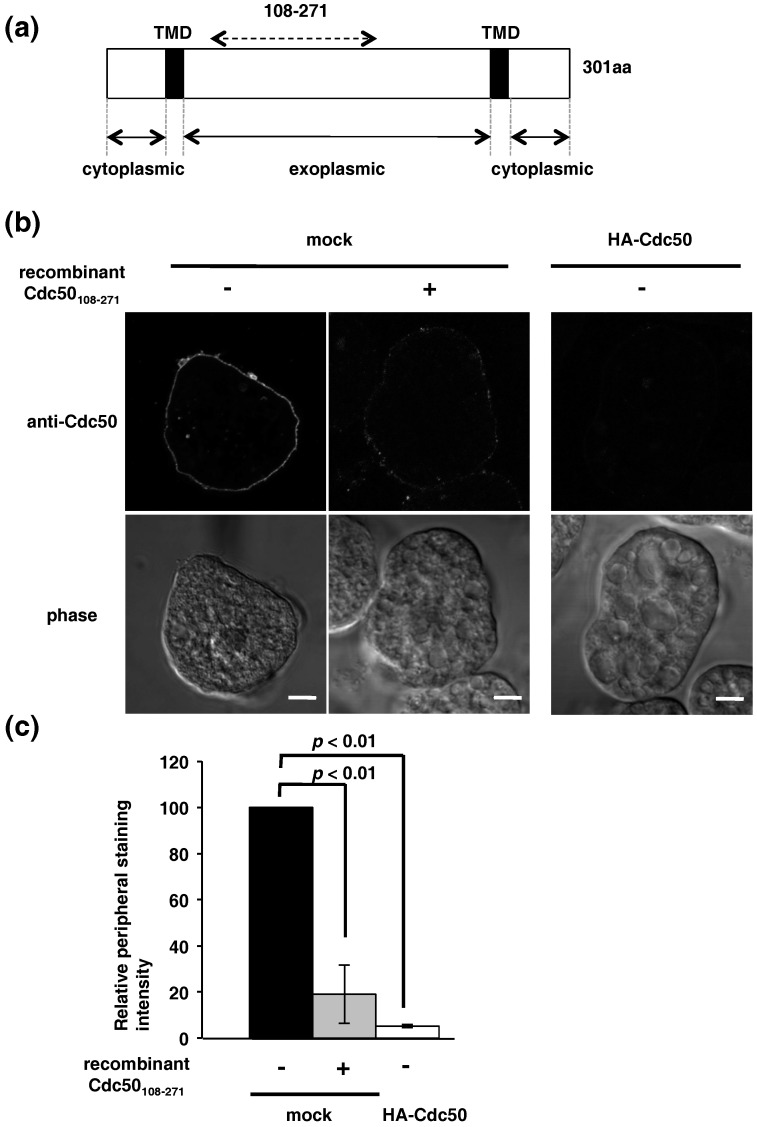
Surface staining of endogenous EhCdc50 in wild-type and overexpressed HA-tagged EhCdc50 cells. (**a**) Schematic diagram of the domain organization of EhCdc50. TMD, transmembrane domain. A part of the exoplasmic domain (a.a. 108–271) that was used to produce *E. coli* recombinant protein to raise antiserum is indicated with a dotted arrow. (**b**) Indirect immunofluorescence assay of endogenous EhCdc50 in mock control and HA-tagged EhCdc50-expressing cells without permeabilization with Triton X100. Pretreatment of the EhCdc50 antiserum with the recombinant protein (middle panel) abolished the surface labeling. Scale bar, 5 µm. (**c**) Quantification of surface labeling with anti-EhCdc50 antibody shown in (**b**). Peripheral signal intensity of 30 independent trophozoites was captured by Zeiss ZEN software. Bar graph shows the means and standard deviations of the relative peripheral fluorescence intensity of EhCdc50 with or without preincubation of the anti-EhCdc50 antibody with recombinant EhCdc50_108-271_ protein at a molar ratio of 1:100 in the mock and HA-EhCdc50 expressing cells, of three independent experiments.

**Figure 4 ijms-19-03831-f004:**
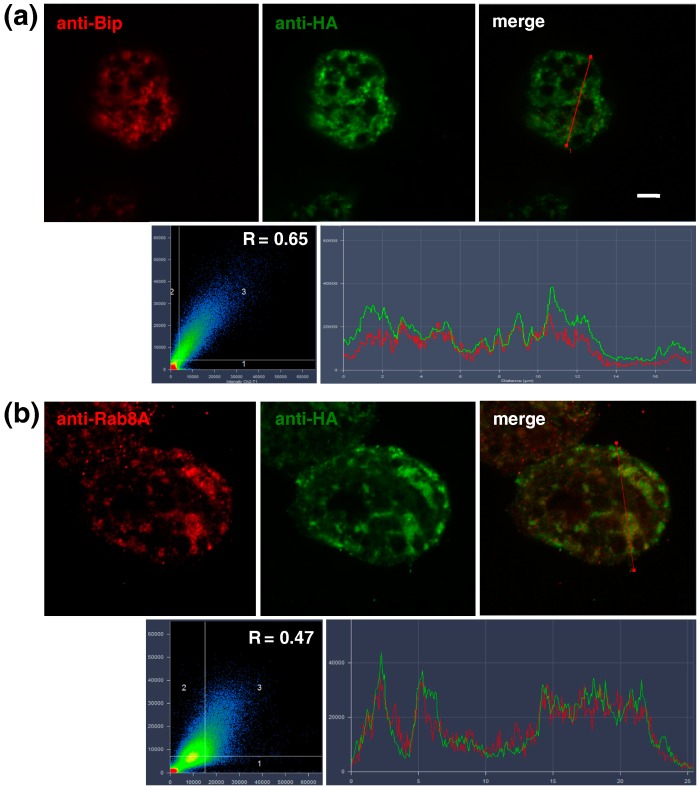
Immunofluorescence assay showing ER localization of overexpressed HA-EhCdc50. HA-EhCdc50 expressing trophozoites were stained with anti-Bip (**a**, red), anti-EhRab8A (**b**, red) and anti-HA (green) antibodies (top panels) after permeabilization with Triton X100. Histograms of the green and red signal intensities along the line indicated in the merged images are shown in the bottom left panels. Scatter plots of colocalization of the two signals in each pixel are shown in the bottom right panels. *R* = Pearson’s correlation coefficient. Scale bar, 5 µm.

**Figure 5 ijms-19-03831-f005:**
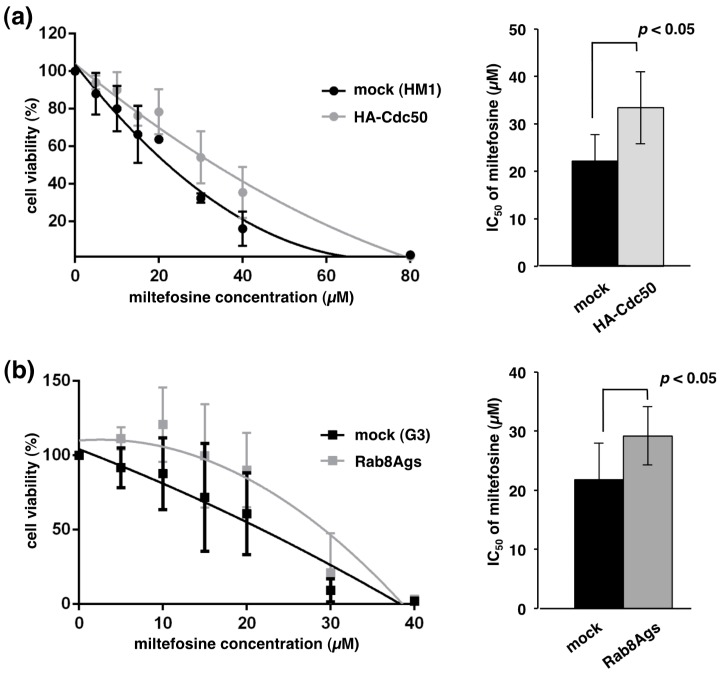
Miltefosine sensitivity of EhCdc50 overexpressing (**a**) and EhRab8A gene silencing cells (**b**). Percentage survival of HA-EhCdc50 expressing line (**a**), EhRab8A gene silenced line (**b**) and its corresponding mock transfected controls of HM-1 (**a**) and G3 strains (**b**) after treatment with indicated concentrations of miltefosine for 18 h. Calculated IC_50_ values using GraphPad Prism ver.6 software are also shown in the right panel. Bar graph shows the means and standard deviations of three independent experiments. The correlation coefficients were calculated using Student’s *t*-test.

**Table 1 ijms-19-03831-t001:** Thirty-five kDa proteins coimmunoprecipitated with Myc-EhRab8A.

Annotation	Gene ID	Molecular Weight (kDa)	Normalized Relative Ratio against Common Peptides, mock/mycRab8A	Subcellular Localization in Other Organisms	E-Value (Species)
Cdc50	EHI_142740	36	0/2.68	trans-Golgi/endosome/PM/ER	1.3 × 10^−43^ (*Arabidopsis thaliana*)
Nuclear pore protein	EHI_118780	38	2.80/9.36	nucleus	5.8 × 10^−18^ (*Chaetomiu thermopholum*)

**Table 2 ijms-19-03831-t002:** Forty kDa proteins coimmunoprecipitated with Myc-EhRab8A.

Annotation	Gene ID	Molecular Weight (kDa)	Normalized Relative Ratio against Common Peptides, mock/mycRab8A	Subcellular Localization in Other Organisms	E-Value (Species)
Sphingomyelinase phosphodiesterase	EHI_100080	46	0/1.02	Acid organella	1.4 × 10^−39^ (*Dictyostelium discoideum*)
tldc domain-containing protein	EHI_134660	42	0/1.01	cytosol	4 × 10^−7^ (*Heterostelium album*)
Vacuolar ATP synthase subunit δ	EHI_106350	40	0/1.01	lysosome membrane	1.3 × 10^−100^ (*Dictyostelium discoideum*)
Sulfate adenylyltransferase	EHI_197160	48	0/1.01	amoebic mitosome lumen	8.8 × 10^−141^ (*Desulfovibrio desulfuricans*)
Glycerophosphodiester phosphodiesterase	EHI_068320	45	0.84/2.02	ER	4.6 × 10^−25^ (*Bacillus subtilis*)
C2 domain containing protein	EHI_069950	37	4.19/9.09	cytosol	5.1 × 10^−7^ (*Arabidopsis thaliana*)

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
