# Peer review of "Identification and Characterization of the Entamoeba Histolytica Rab8a Binding Protein: A Cdc50 Homolog"

_ijms, 2018, doi:10.3390/ijms19123831_

Round 1
Reviewer 1 Report
The title of the manuscript is “Rab8A small GTPase regulates transport of anon-catalytic subunit of phospholipid flippase Cdc50from the endoplasmic reticulum to the plasma membranein the protozoan parasite Entamoeba histolytica”
Rab8A is a small GTPase that regulates the transport of proteins from ER to plasma membrane. Authors identified and characterized a binding partner of EhRab8a through affinity coimmunoprecipitation. Newly identified protein is a Cdc50 homolog, a noncatalytic subunit of lipid flippase, and authors named it as EhCdc50. They also showed by Tunicamycin treatment, endogenously this protein is in N-glycosylated form. The exoplasmic domain of EhCdc50 was developed in E. coliand this recombinant protein was used to produce anti-EhCdc50 in rabbit. Using this antibody authors did immunostaining and showed that Cdc50 localize in plasma membrane. However, trophozoites having overexpressed HA-Cdc50 had significantly decreased peripheral Cdc50, shown by anti-EhCdc50 staining. Authors concluded that overexpression of HA-EhCdc50 inhibit the transport of EhCdc50 from ER to plasma membrane. HA-EhCdc50 overexpressed trophozoites were then permeabilized with Triton X-100 and shown that Cdc50 localize in ER, as anti-BIP (ER marker) and anti-HA colocalized. They also showed, anti-Rab8a colocalize with anti-HA. Finally, authors presented that, silencing of EhRab8a and overexpression of EhCdc50 increased the resistance of trophozoites to miltefosine, an alkylated phosphocoline. It was shown previously in Leishmaniathat, deletion of Cdc50 homolog induce the resistance to miltefosine and this resistance is mediated by reduction of drug incorporation into the cells. Incorporation of miltefosine into the cells requires phospholipid translocation. Authors concluded that silencing of EhRab8a or overexpression of Ha-Cdc50 inhibits the transport of Cdc50 to plasma membrane resulting decreased translocation of phospholipid and hence resistance to miltefosine.
Major comments:
1. In the title authors claimed that Rab8a regulates the transport of Cdc50 from ER to plasma membrane however they did not provide any direct evidence that silencing or deletion of Rab8a inhibit the transport of Cdc50 to plasma membrane. They showed silencing of Rab8a induces miltefosine resistance in trophozoites and reasoned it as a consequence of Cdc50 transport inhibition. I am suggesting following title:
“Identification and Characterization of the Entamoeba histolytica Rab8a binding protein, a Cdc50 homolog”
2. Overexpression of HA-Cdc50 does not inhibit the binding of Rab8a with Cdc50, hence it is not clear why overexpression of HA-Cdc50 inhibits its transport from ER to plasma membrane as it is assumed that Rab8a regulate the transport via interacting with Cdc50.
Minor comments:
1. In figure 1c, it would be better to see immunoblotting using anti-Cdc50 in addition of anti-HA and anti-Rab8a.
2. Figure 5, What test was used to calculate the P value?
3. In para 2.5, it was hypothesized that EhCdc50 are involved in phospholipid translocation. It would have been more informative to measure the phospholipid translocation in EhCdc50 silenced cells.
Author Response
Response to Comments from Reviewer 1
We wish to thank the reviewers for their useful comments. Below are our response to the comments:
Point 1: In the title authors claimed that Rab8a regulates the transport of Cdc50 from ER to plasma membrane however they did not provide any direct evidence that silencing or deletion of Rab8a inhibit the transport of Cdc50 to plasma membrane. They showed silencing of Rab8a induces miltefosine resistance in trophozoites and reasoned it as a consequence of Cdc50 transport inhibition. I am suggesting following title:
“Identification and Characterization of the Entamoeba histolytica Rab8a binding protein, a Cdc50 homolog”
Response 1: We agree that direct evidence of EhRab8-dependent transport of EhCdc50 was not fully provided. We wish to reconfirm that this is the second case of binding protein identification in E. histolytica, which possesses a highly diverse Rab number in the genome and is reportedly important in amoebic virulence. We have revised the title as suggested. The new title is "Identification and Characterization of the Entamoeba histolytica Rab8a binding protein: a Cdc50 homolog"
Point 2: Overexpression of HA-Cdc50 does not inhibit the binding of Rab8a with Cdc50, hence it is not clear why overexpression of HA-Cdc50 inhibits its transport from ER to plasma membrane as it is assumed that Rab8a regulate the transport via interacting with Cdc50.
Response 2: In other organisms, imbalanced expression between Cdc50 and P4-ATPase caused accumulation of Cdc50 together with P4-ATPase to the ER as described in lines 201-203 (Lopez-Marques RL et al., 2010; Saito K et al., 2004; van der Velden LM et al., 2010). We also showed a similar phenotype, that HA-EhCdc50 overexpression caused accumulation of HA-EhCdc50 as well as intrinsic EhCdc50 in the ER, since the surface labeling of Cdc50 disappeared in HA-Cdc50 expressing cells (Fig. 3c). We demonstrated the binding of HA-EhCdc50 and EhRab8A by coimmunoprecipitation (Fig. 2c) and the complex was 87 kDa (Fig. 1a), which was much smaller than ~100 kDa EhP4-ATPases. Thus, we hypothesized that overexpressed EhCdc50 was accumulated in the cargo selection sites in the ER, and under the state of cargo selection by EhRab8A. We described this in the discussion section as follows: "The results of BN-PAGE and immunoblotting showed that EhRab8A formed an 87 kDa complex (Figure 1a), suggesting that the ~110 kDa amoebic EhP4-ATPase, which is present as a family of 11 proteins encoded in the genome (see below), is not part of the EhRab8A/EhCdc50 complex. After EhRab8A recognizes EhCdc50 and completes its sorting in the ER, EhRab8A is presumably replaced by EhP4-ATPase, and then the N-glycosylated EhCdc50/EhP4-ATPase complex can be transported to the plasma membrane." (lines 261- 266).
Minor point 1: In figure 1c, it would be better to see immunoblotting using anti-Cdc50 in addition of anti-HA and anti-Rab8a
Response m1: We also attempted to use anti-Cdc50 antibody. However, the specificity of the anti-Cdc50 rabbit polyclonal antibody was much lower than that of commercially available anti-mouse monoclonal antibodies; hence, it difficult to detect immunoprecipitants using anti-Cdc50 polyclonal antibody.
Minor point 2: Figure 5, What test was used to calculate the P value?
Response m2: We used Student's T-test, and have now included this in the legend in Figure 5 of the revised manuscript.
Minor point 3: In para 2.5, it was hypothesized that EhCdc50 are involved in phospholipid translocation. It would have been more informative to measure the phospholipid translocation in EhCdc50 silenced cells.
Response m3: Yes, we tried to examine lipid flippase activity of EhCdc50 over-expressing cells using fluorescent labeled liposomes as reported in S. cerevisiae cases (Saito K et al., 2004). The lipid flippase activity is an ATP dependent reaction, and in S. cerevisiae, glucose was removed from the medium to establish the glucose-depleted cell condition. We also used the same condition with S. cerevisiae, however, glucose depletion did not effectively provide an ATP-dependent condition in Entamoeba. We could not eliminate the possibility that fluorescent labeled liposomes were incorporated via the endocytosis pathway. Evaluation using miltefosine sensitivity is currently our best system for evaluating the incorporation of lipid analogues.
Reviewer 2 Report
See uploaded document.

Author Response
Response to Comments from Reviewer 2
We wish to thank the reviewers for their useful comments. Below are our response to the comments:
Point 1: Overall, the manuscript is well written and convincingly presented. The solid data is interpreted without pveremphasising results. Some of the images are of poor resolution and I hole this is due to the PDF conversion process and not because of the poor quality of the original files. This needs to be checked though.
Response 1: We have revised Figure 4b to show the co-localization of EhRab8A and HA-EhCdc50. Accordingly, Pearson’s correlation coefficient has now been changed in the revised manuscript (R = 0.47) (line 188).
Point 2: In addition, the authors mention the ER localization of Rab8A in contrast to the more “normal” trans-Golgi localization in other organisms. Perhaps it would aid the reader if the authors mentioned that the Golgi in Entamoeba is perhaps not “normal” either. For example by citing Dacks et al (2003) Pros Biol Sci or Bredeston et al (2005) J Biol Chem. This could perhaps be in line 81.
Response 2: According to the reviewer’s suggestion, we have added the latter reference in line 80 as follows: “However, the function of the Golgi apparatus and the trans-Golgi in Entamoeba is highly diverse in protein glycosylation and organelle function (Bredeston et al., 2005).”
Point 3: Considering the expertise in the host laboratory, I am wondering why the others did not attempt to knock-down Cdc50 as part of the miltefosine experiments. If successful, that would have been more convincing with respect to resistance to miltefosine?
Response 3: We attempted knock-down (phase “gene silencing” in Entamoeba) of EhCdc50, but it was unsuccessful. We have included a phrase to indicate this, as follows: “Gene silencing of EhCdc50 was unsuccessful despite repeated transfections, indicating EhCdc50 is essential for growth.” (lines 157-158).
Minor comments 1: Please indicate glycosylation sites in the yeast homologues (Appendix figure 1)
Response m1: We have now revised Appendix figure 1, and added N-glycosylation sites which are already reported in humans, yeast, and Arabidopsis. The legend was revised as follows: “N-glycosylation sites of Cdc50 homologues reported in humans, S. cerevisiae and A. thaliana, are indicated with small black crosses.” (line 404).
Minor comments 2: Line 298: Perhaps cite Huang et al (2016) Lipid Flippase Subunit Cdc50 Mediates Drug Resistance and Virulence in Cryptococcus neofarmants. mBio 7(3) to refer of miltefosine on Cdc50.
Response m2: We have now referred to Huang et al. (2016), as suggested, in the discussion section as follows: “In the human fungal pathogen Cryptococcus, it is reported that Cdc50 is involved in its virulence as well as in drug resistance” (lines 304-305).